# Transmissibility of COVID-19 depends on the viral load around onset in adult and symptomatic patients

**Hitoshi Kawasuji[1], Yusuke Takegoshi[1], Makito Kaneda[1], Akitoshi Ueno[1], Yuki Miyajima[1], Koyomi Kawago[1], Yasutaka Fukui[1], Yoshihiro Yoshida[2], Miyuki Kimura[2], Hiroshi Yamada[2], Ippei Sakamaki[1], Hideki Tani[2], Yoshitomo Morinaga[2], Yoshihiro Yamamoto[1]***

**1** Department of Clinical Infectious Diseases, Toyama University Graduate School of Medicine and Pharmaceutical Sciences, Toyama, Japan, **2** Department of Microbiology, Toyama University Graduate School of Medicine and Pharmaceutical Sciences, Toyama, Japan

* yamamoto@med.u-toyama.ac.jp

**Data Availability Statement:** All relevant data are within the manuscript.

**Funding:** This study was supported by the Research Program on Emerging and Re-emerging

## Abstract

### Objective

To investigate the relationship between viral load and secondary transmission in novel coronavirus disease 2019 (COVID-19).

### Methods

Epidemiological and clinical data were obtained from immunocompetent laboratory-confirmed patients with COVID-19 who were admitted to and/or from whom viral loads were measured at Toyama University Hospital. Using a case-control approach, index patients who transmitted the disease to at least one other patient were analysed as "cases" (index patients) compared with patients who were not the cause of secondary transmission (non-index patients, analysed as "controls"). The viral load time courses were assessed between the index and non-index symptomatic patients using non-linear regression employing a standard one-phase decay model.

### Results

In total, 28 patients were included in the analysis. Median viral load at the initial sample collection was significantly higher in symptomatic than in asymptomatic patients and in adults than in children. Among symptomatic patients (n = 18), non-linear regression models showed that the estimated viral load at onset was higher in the index than in the non-index patients (median [95% confidence interval]: 6.6 [5.2–8.2] vs. 3.1 [1.5–4.8] log copies/µL, respectively). In adult (symptomatic and asymptomatic) patients (n = 21), median viral load at the initial sample collection was significantly higher in the index than in the non-index patients (p = 0.015, 3.3 vs. 1.8 log copies/µL, respectively).

Infectious Diseases from the Japan Agency for Medical Research and Development (AMED) under Grant Number JP20he0622035.

**Competing interests:** The authors have declared that no competing interests exist.

## Conclusions

High nasopharyngeal viral loads around onset may contribute to secondary transmission of COVID-19. Viral load may help provide a better understanding of why transmission is observed in some instances, but not in others, especially among household contacts.

## Introduction

Coronavirus disease 2019 (COVID-19), which is caused by the novel severe acute respiratory syndrome coronavirus 2 (SARS-CoV-2), has become a global pandemic, and currently threatens human health and lifestyles. Thus, it is important to gain an accurate understanding of the risk of infection with SARS-CoV-2.

The unknown epidemiologic characteristics and transmission dynamics of a novel pathogen, such as SARS-CoV-2, complicate the development and evaluation of effective control policies. In a few recent contact-tracing studies, secondary transmissions were investigated because it gives invaluable clues about transmission dynamics that are more typical [1, 2]. Secondary transmission was defined as the transmission of SARS-CoV-2 from an infected person (source patient) to a secondary patient as ascertained by exposure and symptom onset dates, with no evidence that the secondary patient had been exposed to anyone else with COVID-19.

A few preliminary contact-tracing studies showed that the highest-risk exposure setting of COVID-19 transmission was the household [2]. Nevertheless, it is not known when and how long a patient with COVID-19 should be isolated or whether close contacts should be quarantined. Additional information is needed about the transmission risk.

In addition, some case reports and modeling studies suggest asymptomatic carriage of SARS-CoV-2 plays a role in transmission [3]. Studies have shown that 17.9–19.2% of SARS-CoV-2 infections are asymptomatic [4, 5], which poses tremendous infection control challenges.

Generally, the risk of microbial human-to-human transmission is dependent on the duration of the highly-infectious phase and the number of virions contained in air particulates such as droplets and aerosols. However, as it is a new disease, little is yet known about COVID-19 and the risk of infection in various situations.

The viral load of SARS-CoV-2 peaks around the time of symptom onset, followed by a gradual decrease to a low level after about 10 days [6, 7]. Regarding the period of high infectiousness, a recent study reported that exposure to an index case within 5 days of symptom onset confers a high risk of secondary transmission [2]. The high transmissibility around symptom onset gradually decreases, consistent with the dynamical pattern of viral shedding [8].

In addition, viral load can be associated with infectiousness, especially in the acute phase of COVID-19. However, little information is available on the relationship between SARS-CoV-2 viral load in nasopharyngeal swab specimens (nasopharyngeal viral load), which are usually obtained for serial viral load monitoring, and secondary transmission. We hypothesized that high nasopharyngeal viral loads contribute to secondary transmission of COVID-19 and viral loads may be higher among cases who transmit to others compared to cases who do not transmit to others. In this study, we reviewed patients with COVID-19, including family clusters, and conducted follow-up interviews to investigate the relationship between viral load and secondary infection.

## Materials and methods

### Study design and participants

Epidemiological and clinical data were obtained from immunocompetent laboratory-confirmed patients with COVID-19 who were admitted to and/or from whom viral loads were measured at Toyama University Hospital from April 13 to May 7, 2020. The patients were divided into two groups: those who subsequently transmitted the disease to at least one other patient (index patients), and those who were not the cause of secondary transmission (non-index patients). Index patients and infected contacts were confirmed by positive results of their nasopharyngeal swab samples on SARS-CoV-2 quantitative reverse transcriptase–polymerase chain reaction (RT-qPCR), and uninfected close contacts were those who were at least once tested negative on SARS-CoV-2 RT-qPCR. Index patients and those with secondary transmission were estimated based on serial intervals in the family clusters. Epidemiological and clinical data, including the number of close contacts and secondary patients in health care, household, or other social settings, were investigated through structured telephone interviews with the patients and their families, and based on available information from public health centers.

### Data collection

For each patient, the following data were retrieved from medical charts and structured telephone interview sheets (S1 and S2 Appendices): demographics, clinical presentation, date of symptom onset, exposure history in the 14 days before symptom onset, date of initial sample collection, need for supplemental oxygen (moderate) and/or mechanical ventilation (severe), and dates of the first negative RT-qPCR test result and hospital discharge. The potential confounding factors which may have modified the observed viral load, such as received treatment including combinations of antivirals and antibiotics, have not been systematically investigated. We conducted the structured telephone interviews and accessed the medical records from May 18–22, 2020 to obtain the data used in the present study.

### RT-qPCR

Nasal swab specimens were pretreated with 500 μL of Sputazyme (Kyokuto Pharmaceutical, Tokyo, Japan). After centrifugation at 20,000 × g for 30 min at 4˚C, the supernatant was used for RNA extraction. A total of 60 μL of RNA solution was obtained from 140 μL of the supernatant using the QIAamp ViralRNA Mini Kit (QIAGEN, Hilden, Germany) or Nippongene Isospin RNA Virus (Nippongene, Tokyo, Japan) according to the manufacturer's instructions. The viral loads of SARS-CoV-2 were quantified based on an N2-gene-specific primer/probe set by RT-qPCR according to the Japan National Institute of Infectious Diseases protocol [9]. The quality of quantification was controlled by AcroMetrix Coronavirus 2019 (COVID-19) RNA Control (Thermo Fisher Scientific, Fremont, CA). The detection limit was approximately 0.4 copies/μL (2 copies/5 μL).

### Statistical analysis

Continuous and categorical variables were presented as the median (interquartile range [IQR]) and n (%), respectively. We used the Mann–Whitney U test, $\chi^2$ test, or Fisher's exact test to compare differences between the index and non-index patients where appropriate. Data were analyzed using JMP Pro version 14.2.0 software (SAS Institute Inc., Cary, NC, USA). The viral load time courses were assessed using nonlinear regression employing a standard one-phase decay model in Prism version 8.4.2 (GraphPad Software Inc., San Diego, CA, USA).

### Ethics approval

The study was performed in conformity with the Helsinki Declaration, after approval by the Ethical Review Board of University of Toyama (approval number: R2019167). Written informed consent was obtained from all patients.

## Results

Among the 28 patients (median age, 45.5 years) with laboratory-confirmed COVID-19, 15 (53.6%) were male, 21 (75.0%) were adults (18 years or older), and 10 (35.7%) were asymptomatic (Table 1). Among their 105 close contacts, 14 paired index-secondary cases were found. Fourteen (50.0%) were index patients within 11 family clusters, 10 (35.7%) were secondary transmission patients without further spreading within seven family clusters, and the remaining four (14.3%) were sporadic cases. Of the 18 symptomatic patients, the numbers of mild, moderate, and severe cases were 12, 5, and 1, respectively.

A total of 89 nasopharyngeal swabs were collected from the 28 patients from 2 to 36 days after onset. The median (IQR) time at initial sample collection was 6 (2.8–9) days after symptom onset. The median viral load at the initial sample collection was significantly higher in adults than in children (p = 0.02, 2.3 vs. 0.9 log copies/μL, respectively); however, viral loads during follow-up were not significantly different between the two groups (p = 0.89). In addition, the median viral load at initial sample collection was significantly higher in symptomatic than in asymptomatic patients (p<0.01, 2.8 vs. 0.9 log copies/μL, respectively).

Next, the viral load in symptomatic patients was compared between the index and non-index patients (Table 2). Viral loads peaked soon after symptom onset, and then gradually decreased toward the detection limit (Fig 1A). The time to viral clearance from onset in the index patients was 21 (15–31) days (median [range]), and no significant difference was found between the index and non-index patients (p = 0.34, 17 [9–26] days, median [range]).

**Table 1. Basic characteristics of the patients.**

| Characteristics | Study population |
|---|---|
| Age, median, y | 45.5 |
| 0–17, n (%) | 7 (25.0) |
| 18–64, n (%) | 14 (50.0) |
| ≥65, n (%) | 7 (25.0) |
| Sex | |
| Male, n (%) | 15 (53.6) |
| Situation, n (%) | |
| Index | 14 (50.0) |
| 0–17, n (%) | 4 (14.3) |
| 18–64, n (%) | 6 (21.4) |
| ≥65, n (%) | 4 (14.3) |
| Secondary | 10 (35.7) |
| Sporadic | 4 (14.3) |
| Presence of symptoms, n (%) | |
| Asymptomatic | 10 (35.7) |
| Symptomatic | 18 (64.3) |
| Mild | 12 (42.9) |
| Moderate | 5 (17.9) |
| Severe | 1 (3.6) |

**Table 2. Summary of viral load in the index and non-index symptomatic patients.**

| Characteristics | Index patients, n = 11 | Non-index patients, n = 7 | P-value |
|---|---|---|---|
| Initial sampling | | | |
| Sampling days after onset, median (range) | 6 (2 to 12) | 4 (−1 to 10) | 0.52 |
| Viral load (log copies/μL), median (range) | 3.1 (1.6 to 5.2) | 1.9 (−0.4 to 4.6) | 0.15 |
| Trend of viral load (log copies/μL), median (range) | | | |
| −1 to 5 days after onset | 4.9 (4.4 to 5.2) | 3.0 (0.7 to 4.6) | 0.11 |
| 6 to 10 days after onset | 2.3 (1.6 to 4.9) | 0.7 (−0.4 to 3.1) | 0.17 |
| Days to viral clearance from onset, median (range) | 21 (15 to 31) | 17 (9 to 26) | 0.34 |

Among the symptomatic patients, the nasopharyngeal viral loads at the initial sample collection were not significantly different between the index and non-index patients (p = 0.15, median [range]: 3.1 [1.6 to 5.2] vs. 1.9 [−0.4 to 4.6] log copies/μL, respectively). However, non-linear regression models using all the data from the index or non-index patients showed that the viral load of the index patients at onset was higher than that of the non-index patients (median [95% confidence interval]: 6.6 [5.2 to 8.2] vs. 3.1 [1.5 to 4.8] log copies/μL, respectively), and this trend continued until 10 days after onset (Fig 1B).

In addition, in adult (symptomatic and asymptomatic) patients, the viral load was also compared between the index and non-index patients. Although the time to initial sampling after onset and the proportion of asymptomatic patients were not significantly different between the two groups, the nasopharyngeal viral loads at the initial sample collection were significantly higher in the index patients than in the non-index patients (p = 0.015, median [range]: 3.3 [1.6 to 5.2] vs. 1.8 [-0.4 to 4.6] log copies/μL, respectively).

## Discussion

In this study, we analyzed viral loads in nasopharyngeal swabs obtained from 18 symptomatic and 10 asymptomatic patients with laboratory-confirmed COVID-19 to assess the relationship between viral load and secondary transmission.

Although a recent study suggested that the viral load detected in asymptomatic patients was similar to that in symptomatic patients [7], in this study, the viral load at the time of initial

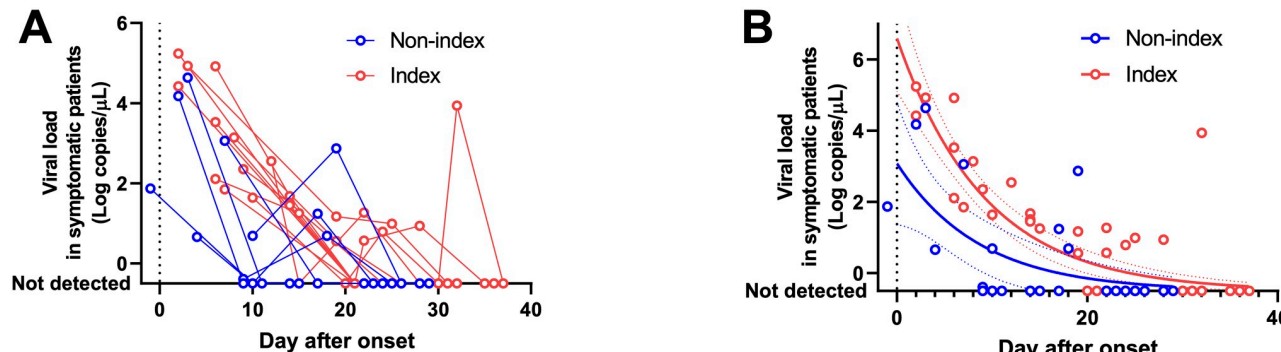

**Fig 1. Trends in viral loads in the symptomatic patients.** (A) Viral load time courses of index (red) and non-index patients (blue). (B) Nonlinear regression models of index (red) and non-index patients (blue). The models were calculated by using all the data of the index or non-index patients. When no virus was detected, the data were hypothetically plotted as −0.5 log copies/μL. Solid curves are best-fit models and dotted lines indicate the 95% confidence interval of each model.

sample collection was significantly higher in symptomatic than in asymptomatic patients. In addition, we also found that the viral load at the time of initial sample collection was significantly higher in adult than in children.

Furthermore, among the adult and the symptomatic patients, the viral loads in the index patients were significantly higher than those in the non-index patients. Further studies are needed to confirm these results, as this study only involved a small number of patients. However, it is plausible that high nasopharyngeal viral loads contribute to secondary transmission of COVID-19. To our knowledge, no previous study has assessed the relationship between viral load and secondary transmission. Among the asymptomatic carriers, especially children, it was difficult to determine whether the viral load could impact transmission because no carrier had a high viral load. However, a recent study reported that persons with asymptomatic infections appeared to be less effective in transmitting the virus [1]. Although viral load levels at the initial sample collection may not have been directly comparable between symptomatic and asymptomatic patients because the data collection period may differ, our results may partially explain the difference of secondary attack rate of COVID-19 between in symptomatic patients and asymptomatic patients. However, this finding should not discourage isolation and surveillance efforts.

After symptom onset, the viral load decreased monotonically. The median time to clearance of the virus was similar to that in a previous report, in which virus was detected for a median of 20 days (up to 37 days among survivors) after symptom onset [10]. However, the median time did not differ between the index and non-index patients. These findings imply that viral clearance is independent of the initial virus load, but may be regulated by the host immune response. In most patients, the virus was detected for about 3 weeks; however, when the risk of transmission disappears remains unclear. A previous report demonstrated that infectiousness may decline at 8 days after symptom onset [11]. Although the relationship between viral load and the infectiousness of COVID-19 remains unknown [12], the present study provides insight into the viral load threshold associated with infectivity.

This study has several limitations inherent to the small sample size and potential for confounding viral load and clinical conditions that cannot be excluded. Also, we could not perform case-control matching due to the small number of patients. The date of symptom onset and disappearance and information on disease transmission relied on self-reported information from the patients and their families, as well as available information from public health centers, which could potentially lead to missing some cases of secondary transmission. In addition, the viral load dynamics were based on data from patients who received treatment, including combinations of antivirals and antibiotics, which could have modified the patterns of the viral load dynamics.

During the COVID-19 pandemic, better understanding of the relationship between viral load and secondary transmission is important for the development and evaluation of effective control policies. Although it is not known when and how long a patient with COVID-19 should be isolated or whether close contacts should be quarantined, our results suggested that viral load may help the decision to when to discharge isolation, how wide the range of close-contact tracing is needed in individual patients.

In conclusion, the results of this study suggest that a high nasopharyngeal viral load may contribute to the secondary transmission of COVID-19. In addition, the viral load may help explain why transmission is observed in some instances, but not in others, especially among household contacts. Although RT-qPCR does no distinguish between infectious virus and noninfectious nucleic acid, our findings may lead to the establishment of a viral load threshold to clarify COVID-19 disease transmission and infectivity.

## Supporting information

**S1 Appendix. The telephone interview questionnaire in English.**
(DOCX)

**S2 Appendix. The telephone interview questionnaire in the original language.**
(DOCX)

**S1 Data.**
(XLSX)

## Acknowledgments

We thank Ai Kakumoto for supporting PCR.

## Author Contributions

**Conceptualization:** Hitoshi Kawasuji, Yusuke Takegoshi, Yoshitomo Morinaga, Yoshihiro Yamamoto.

**Data curation:** Hitoshi Kawasuji, Yoshihiro Yoshida, Miyuki Kimura, Hiroshi Yamada, Hideki Tani, Yoshitomo Morinaga.

**Formal analysis:** Hitoshi Kawasuji, Yoshitomo Morinaga.

**Investigation:** Hitoshi Kawasuji, Yusuke Takegoshi, Akitoshi Ueno, Yuki Miyajima, Yasutaka Fukui, Yoshihiro Yoshida, Miyuki Kimura, Hiroshi Yamada, Hideki Tani.

**Methodology:** Hitoshi Kawasuji, Yoshitomo Morinaga, Yoshihiro Yamamoto.

**Project administration:** Hitoshi Kawasuji, Yusuke Takegoshi, Yoshitomo Morinaga, Yoshihiro Yamamoto.

**Resources:** Yoshihiro Yoshida, Miyuki Kimura, Hiroshi Yamada, Hideki Tani, Yoshitomo Morinaga.

**Supervision:** Makito Kaneda, Akitoshi Ueno, Yuki Miyajima, Koyomi Kawago, Yasutaka Fukui, Ippei Sakamaki, Hideki Tani, Yoshitomo Morinaga, Yoshihiro Yamamoto.

**Validation:** Ippei Sakamaki, Yoshihiro Yamamoto.

**Visualization:** Yoshihiro Yamamoto.

**Writing – original draft:** Hitoshi Kawasuji, Yoshitomo Morinaga.

**Writing – review & editing:** Yoshitomo Morinaga, Yoshihiro Yamamoto.

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
