## [Decision Letter · Decision Letter 0]

10 Aug 2020

PONE-D-20-20259

Transmissibility of COVID-19 depends on the viral load around onset in adult and symptomatic patients

PLOS ONE

Dear Dr. Yamamoto,

Thank you for submitting your manuscript to PLOS ONE. After careful consideration, we feel that it has merit but does not fully meet PLOS ONE’s publication criteria as it currently stands. Therefore, we invite you to submit a revised version of the manuscript that addresses the points raised during the review process. Specifically, please address the following areas.

1) Introduction: Define secondary transmission and clarify its importance to the topic (COVID-19). Clarify your research question and hypothesis. Clarify nasopharyngeal viral load and be consistent with the term through the manuscript. 

2) Method: Clarify the case-control matching process (details are needed such as how the cases were matched with controls) and a graph of matching is needed. 

3)Discussion: Discuss the implementation of findings of the study in Epidemiology and Public Health. 

We look forward to receiving your revised manuscript.

Kind regards,

Xi Pan

Academic Editor

PLOS ONE

Journal Requirements:

2. Please include additional information regarding the telephone interview guide used in the study and ensure that you have provided sufficient details that others could replicate the analyses. For instance, if you developed the telephone questionnaire as part of this study and it is not under a copyright more restrictive than CC-BY, please include a copy, in both the original language and English, as Supporting Information.

3. Please include the date(s) on which you accessed the databases or records to obtain the data used in your study.

Reviewers' comments:

Reviewer's Responses to Questions

**Comments to the Author**

1. Is the manuscript technically sound, and do the data support the conclusions?

Reviewer #1: Yes

2. Has the statistical analysis been performed appropriately and rigorously? 

Reviewer #1: Yes

3. Have the authors made all data underlying the findings in their manuscript fully available?

Reviewer #1: Yes

4. Is the manuscript presented in an intelligible fashion and written in standard English?

Reviewer #1: Yes

5. Review Comments to the Author

Reviewer #1: The article is well written and the information provided is of relevance in the present scenario of the pandemic. The study design is also suitable and the statistical analysis performed is also accurate. However, it is advised that the potential confounding factors are clearly mentioned in the methods section.

The introduction can be further strengthened by using the following references and also be utilized in the discussion component:

Mizumoto K, Kagaya K, Zarebski A, Chowell G. Estimating the asymptomatic proportion of coronavirus disease 2019 (COVID-19) cases on board the Diamond Princess cruise ship, Yokohama, Japan, 2020. Eurosurveillance. 2020 Mar 12;25(10):2000180.

Kim GU, Kim MJ, Ra SH, Lee J, Bae S, Jung J, Kim SH. Clinical characteristics of asymptomatic and symptomatic patients with mild COVID-19. Clinical Microbiology and Infection. 2020 May 1.

Bai Y, Yao L, Wei T, Tian F, Jin DY, Chen L, Wang M. Presumed asymptomatic carrier transmission of COVID-19. Jama. 2020 Apr 14;323(14):1406-7.

6. PLOS authors have the option to publish the peer review history of their article (what does this mean?). If published, this will include your full peer review and any attached files.

Reviewer #1: No

---

## [Author Response · Author response to Decision Letter 0]

9 Oct 2020

Editor (Comments for the Author):

Specific Comments:

1) Introduction: Define secondary transmission and clarify its importance to the topic (COVID-19). Clarify your research question and hypothesis. Clarify nasopharyngeal viral load and be consistent with the term through the manuscript.

Response: We added the following sentence as suggested in the introduction section: 

“The unknown epidemiologic characteristics and transmission dynamics of a novel pathogen, such as SARS-CoV-2, complicate the development and evaluation of effective control policies. In a few recent contact-tracing studies, secondary transmissions were investigated because it gives invaluable clues about transmission dynamics that are more typical [1,2]. Secondary transmission was defined as the transmission of SARS-CoV-2 from an infected person (source patient) to a secondary patient as ascertained by exposure and symptom onset dates, with no evidence that the secondary patient had been exposed to anyone else with COVID-19. 

A few preliminary contact-tracing studies showed that the highest-risk exposure setting of COVID-19 transmission was the household [2]. Nevertheless, it is not known when and how long a patient with COVID-19 should be isolated or whether close contacts should be quarantined. Additional information is needed about the transmission risk.” (Lines 52-63 in the Manuscript, Lines 52-63 in the Revised Manuscript with Track Changes).

“In addition, viral load can be associated with infectiousness, especially in the acute phase of COVID-19. However, little information is available on the relationship between SARS-CoV-2 nasopharyngeal viral load in nasopharyngeal swab specimens (nasopharyngeal viral load), which are usually obtained for serial viral load monitoring, and secondary transmission. We hypothesized that high nasopharyngeal viral loads contribute to secondary transmission of COVID-19 and viral loads may be higher among cases who transmit to others compared to cases who do not transmit to others. In this study, we reviewed patients with COVID-19, including family clusters, and conducted follow-up interviews to investigate the relationship between viral load and secondary infection.” (Lines 77-85 in the Manuscript, Lines 77-85 in the Revised Manuscript with Track Changes).

2) Method: Clarify the case-control matching process (details are needed such as how the cases were matched with controls) and a graph of matching is needed.

Response: Although all immunocompetent patients who were admitted to and/or from whom viral loads were measured at Toyama University Hospital from April 13 to May 7, 2020 were included in the present study, it was difficult to use matched case-control design due to the small number of patients. Nevertheless, there was no significantly difference in patients’ demographics and clinical presentation between the index and non-index symptomatic patients.

We added the following sentence: “Also, we could not perform case-control matching due to the small number of patients.” (Lines 213-214 in the Manuscript, Lines 213-214 in the Revised Manuscript with Track Changes). 

3) Discussion: Discuss the implementation of findings of the study in Epidemiology and Public Health.

Response: We added the following sentence as suggested: “During the COVID-19 pandemic, better understanding of the relationship between viral load and secondary transmission is important for the development and evaluation of effective control policies. Although it is not known when and how long a patient with COVID-19 should be isolated or whether close contacts should be quarantined, our results suggested that viral load may help the decision to when to discharge isolation, how wide the range of close-contact tracing is needed in individual patients.” (Lines 221-226 in the Manuscript, Lines 221-226 in the Revised Manuscript with Track Changes).

Journal Requirements:

2. Please include additional information regarding the telephone interview guide used in the study and ensure that you have provided sufficient details that others could replicate the analyses. For instance, if you developed the telephone interview questionnaire as part of this study and it is not under a copyright more restrictive than CC-BY, please include a copy, in both the original language and English, as Supporting Information.

Response: We added telephone interview questionnaire in both English and the original language as Supporting Information (S1 Appendix and S2 Appendix). 

We added the following sentence:

“For each patient, the following data were retrieved from medical charts and structured telephone interview sheets (S1 Appendix and S2 Appendix): demographics, clinical presentation, date of symptom onset, exposure history in the 14 days before symptom onset, date of initial sample collection, need for supplemental oxygen (moderate) and/or mechanical ventilation (severe), and dates of the first negative RT-qPCR test result and hospital discharge.” (Lines 103-107 in the Manuscript, Lines 103-107 in the Revised Manuscript with Track Changes).

“Supporting Information

S1 Appendix. The telephone interview questionnaire in English.

S2 Appendix. The telephone interview questionnaire in the original language.”

(Lines 234-236 in the Manuscript, Lines 234-236 in the Revised Manuscript with Track Changes).

3. Please include the date(s) on which you accessed the databases or records to obtain the data used in your study.

Response: We added the following sentence: 

“Epidemiological and clinical data were obtained from immunocompetent laboratory-confirmed patients with COVID-19 who were admitted to and/or from whom viral loads were measured at Toyama University Hospital from April 13 to May 7, 2020.” (Lines 88-90 in the Manuscript, Lines 88-90 in the Revised Manuscript with Track Changes).

“We conducted the structured telephone interviews and accessed the medical records from May 18-22, 2020 to obtain the data used in the present study.” (Lines 110-111 in the Manuscript, Lines 110-111 in the Revised Manuscript with Track Changes).

Reviewer #1 (Comments for the Author):

The article is well written and the information provided is of relevance in the present scenario of the pandemic. The study design is also suitable and the statistical analysis performed is also accurate. However, it is advised that the potential confounding factors are clearly mentioned in the methods section.

Response: We agree with the reviewer’s comment. We added the following sentence:

“The potential confounding factors which may have modified the observed viral load, such as received treatment including combinations of antivirals and antibiotics, have not been systematically investigated.” (Lines 108-110 in the Manuscript, Lines 108-110 in the Revised Manuscript with Track Changes).

The introduction can be further strengthened by using the following references and also be utilized in the discussion component:

Mizumoto K, Kagaya K, Zarebski A, Chowell G. Estimating the asymptomatic proportion of coronavirus disease 2019 (COVID-19) cases on board the Diamond Princess cruise ship, Yokohama, Japan, 2020. Eurosurveillance. 2020 Mar 12;25(10):2000180.

Kim GU, Kim MJ, Ra SH, Lee J, Bae S, Jung J, Kim SH. Clinical characteristics of asymptomatic and symptomatic patients with mild COVID-19. Clinical Microbiology and Infection. 2020 May 1.

Bai Y, Yao L, Wei T, Tian F, Jin DY, Chen L, Wang M. Presumed asymptomatic carrier transmission of COVID-19. Jama. 2020 Apr 14;323(14):1406-7.

Response: We agree with the reviewer’s comment. We added the following sentences in the introduction and discussion sections:

“In addition, some case reports and modeling studies suggest asymptomatic carriage of SARS-CoV-2 plays a role in transmission [3]. Studies have shown that 17.9-19.2% of SARS-CoV-2 infections are asymptomatic [4,5], which poses tremendous infection control challenges.” (Lines 64-66 in the Manuscript, Lines 64-66 in the Revised Manuscript with Track Changes).

“However, a recent study reported that persons with asymptomatic infections appeared to be less effective in transmitting the virus [1]. Although viral load levels at the initial sample collection may not have been directly comparable between symptomatic and asymptomatic patients because the data collection period may differ, our result may partially explain the difference of secondary attack rate of COVID-19 between in symptomatic patients and asymptomatic patients. However, this finding should not discourage isolation and surveillance efforts.” (Lines 195-201 in the Manuscript, Lines 195-201 in the Revised Manuscript with Track Changes).

---

## [Decision Letter · Decision Letter 1]

25 Nov 2020

Transmissibility of COVID-19 depends on the viral load around onset in adult and symptomatic patients

PONE-D-20-20259R1

Dear Dr. Yamamoto,

We’re pleased to inform you that your manuscript has been judged scientifically suitable for publication and will be formally accepted for publication once it meets all outstanding technical requirements.

Kind regards,

Xi Pan

Academic Editor

PLOS ONE

Additional Editor Comments (optional):

Reviewers' comments:

Reviewer's Responses to Questions

**Comments to the Author**

1. If the authors have adequately addressed your comments raised in a previous round of review and you feel that this manuscript is now acceptable for publication, you may indicate that here to bypass the “Comments to the Author” section, enter your conflict of interest statement in the “Confidential to Editor” section, and submit your "Accept" recommendation.

Reviewer #1: All comments have been addressed

2. Is the manuscript technically sound, and do the data support the conclusions?

Reviewer #1: Yes

3. Has the statistical analysis been performed appropriately and rigorously? 

Reviewer #1: Yes

4. Have the authors made all data underlying the findings in their manuscript fully available?

Reviewer #1: Yes

5. Is the manuscript presented in an intelligible fashion and written in standard English?

Reviewer #1: Yes

6. Review Comments to the Author

Reviewer #1: Thank you for addressing all the comments that were previously highlighted in the detailed review. The manuscript is now suitable for publication.

7. PLOS authors have the option to publish the peer review history of their article (what does this mean?). If published, this will include your full peer review and any attached files.

Reviewer #1: No

---

## [Editor Report · Acceptance letter]

1 Dec 2020

PONE-D-20-20259R1 

Transmissibility of COVID-19 depends on the viral load around onset in adult and symptomatic patients 

Dear Dr. Yamamoto:

I'm pleased to inform you that your manuscript has been deemed suitable for publication in PLOS ONE. Congratulations! Your manuscript is now with our production department. 

Kind regards, 

on behalf of

Dr. Xi Pan 

Academic Editor

PLOS ONE